# Neurofeedback Technology Reduces Cortisol Levels in Bruxismitle Patients: Assessment of Cerebral Activity and Anxiolytic Effects of *Origanum majorana* Essential Oil

**DOI:** 10.3390/biomimetics9110715

**Published:** 2024-11-20

**Authors:** José Joaquín Merino, José María Parmigiani-Izquierdo, Adolfo Toledano Gasca, María Eugenia Cabaña-Muñoz

**Affiliations:** 1Dpto. Farmacología, Farmacognosia y Botánica, Facultad de Farmacia, Universidad Complutense de Madrid (UCM), 28040 Madrid, Spain; 2CIROM Center, 30001 Murcía, Spain; jmparmi@clinicacirom.com (J.M.P.-I.); mecjj@clinicacirom.com (M.E.C.-M.); 3Cajal Institute, 28002 Madrid, Spain; atoledano@cajal.csic.es

**Keywords:** cortisol, NeurOptimal/neurofeedback, neural plasticity, brain stimulation, aromatherapy (*Origanum majorana* essential oil), nutritional supplementation, neuromodulation, stress, brain activity

## Abstract

Cerebral activities were measured during 21 essions in NeurOptimal (NO)-trained patients with bruxism. Salivary cortisol levels were quantified for each six training sessions (session 1, 6, 12, 18, 21) in 12 patients with bruxism after performing their pre- and post-NeurOptimal sessions. Their cortisol levels were compared with controls (without stress). We evaluated whether NO overtraining could reduce stress in bruxism after 21 repeated sessions with/without *Origanum majorana* inhalation by using nasal impregned filters with this essential oil (*n* = 12). This study enrolled 89 participants (590 salivary samples for cortisol assessment by ELISA ng/mL). Salivary samples were collected at several NO learning sessions (session 1, 6, 12, 18, and 21). In the present study, we assessed whether *Origanum majorana* essential oil exposure during 21 NO training sessions can promote anxiolytic effects by reducing cortisol levels in Bruxismitle patients or modulate their brain activities. The experimental design also included control subjects without NO training (*n* = 30) and unstressed participants without bruxism, as well as trained NeurOptimal (*n* = 5) participants during the 21 sessions, also including control subjects without stress. In our study, NeurOptimal post-training decreased cortisol levels in Bruxismitle patients, reducing stress scores on the Hamilton II scale after 21 NO sessions; finally, *Origanum majorana* essential oil exposure during NO training could enhance anxiolytic effects of repeated NO in Bruxismitle patients. The parameter divergence as an index of cerebral activity evaluates the reached difference between cerebral activity at pre-learning (PRE) minus post-training (POST) values in Bruxismitle participants with/without *Origanum majorana* odor exposure during each NO training sessions. As a consequence of NO overtraining, these cerebral activities fluctuate reaching a calm state while anxious states are associated with high divergences. The reduction in divergences when they are close to zero by habituation means a final calm state is reached by NO overtraining, while higher divergences mean anxiogenic states. Collectively, *Origanum majorana* essential oil inhalation during NO training could decrease salivary cortisol levels after 21 NO training sessions in Bruxismitle.

## 1. Introduction

It is known that the stress hormone, cortisol, is released by stressful conditions, anxiety, or depression in patients [1]. HPA (hypothalamic adrenal activity) regulation under stress also modulates learning experiences [2,3], while the persistence of stress leads to neuroadaptations in prefrontal and limbic areas of patients with stress [4,5,6]. Anxiety disorders affect 40 million people in the USA and contribute to the progression of certain neuropsychiatric diseases [7]. The inhalation of different odors can affect emotional responses in humans [8,9]. In this context, since certain essential oils can induce antioxidant or anti-inflammatory effects, we have studied the potential anxiolytic effect of *Origanum majorana* essential oil and/or during NeurOptimal overtraining in Bruxismitle participants. The *Origanum dictamnus* essential oil has some constituents, such as carvacrol (52%) and gamma-terpinene (8.4%), that contribute to its antioxidant properties [10]. The *Origanum majorana* essential oil used in the present study has a composition based in terpenoid oils (alpha-terpinene 14.10%, gamma terpinene 14.1%, *cis*-Tuyhanol 15.2%) and other minority products.

On other hand, neurofeedback is a noninvasive self-training regimen that improves brain function in the prefrontal cortex, the cingulate cortex [11], by regulating waves in specific brain areas as a consequence of overtraining [12,13]. NeurOptimal (NO, Zengar Institute, Canada) is a version of neurofeedback neurotechnology based on the training of patients by listening to a piece of new age music (the same for all of them), given its beneficial effects on several physiological parameters [14,15,16,17,18]. The collective recording of neuron electrical activity generates brain waves that constitutes electroencephalography (EEG) [19]. During NO training sessions, the electrodes are placed in the prefrontal area of patients in order to record changes in cerebral brain activity when disruptions in music listening, “music interruptions”, lead to changes in wave perceptions (https://neuroptimal.com (accessed on 7 June 2020)). Local and inhibitory/excitatory interactions shape neuronal representations of sensory, motor, and cognitive variables, and produce local electroencephalographic (EEG) gamma frequency (30–80 Hz) oscillations [19,20]. Alpha and beta wave activities may be used in different ways for detecting emotional changes in patients [21]. In fact, EEG contains several waves depending on each frequency; for instance, delta waves (0.5~3 Hz), theta waves (4~7 Hz), alpha waves (8~12 Hz), SMR (sensorimotor rhythm) waves (13~15 Hz), low beta waves (16~20 Hz), and high beta waves (21~40 Hz) are associated with different emotions in patients. In this way, a fast wave is associated with high arousal, concentration, and focused attention in neurofeedback trained participants [22] After neurofeedback training, participants become capable of modulating their brain wave activity in real time with specially designed computer equipment [12,13,23,24], as also happens in NeurOptimal (NO) training. Thus, neurofeedback training, including NO training, induces a beneficial effect on working memory, attention [25], and cognitive processes [13,26,27,28,29,30]. Until now, all neurofeedback-based research studies on emotion were evaluated in healthy adults and in patients with brain disabilities [31,32,33,34,35,36,37].

Bruxism is characterized as repetitive jaw muscle activity by clenching or grinding the teeth and/or bracing or thrusting of the mandible. The correlation between cortisol, bruxism and the etiopathogenesis of temporomandibular diseases have been confirmed by a meta-analysis that included studies from 2008 to 2020 [27].

### Aim

To study whether NeurOptimal (NO) overtraining during 21 sessions could decrease cerebral activities (Divergence) by measuring brain activities at pre-training minus post-training values in Bruxismitle patients.

To study whether *Origanum majorana* essential oil impregnation on nasal filters could reduce cortisol levels during 21 NO training sessions in Bruxismitle patients at certain pre-/post-training session ((S1), 6, 12, 18, 21).

## 2. Materials and Methods

### 2.1. Origanum Majorana Essential Oil Composition

*Origanum majorana* essential oil (Pranarom, Barcelona, Spain) was directly impregned at 1% (diluted with neutral essential oil) on nasal filters of active carbon (InspiraHealth, Barcelona, Spain). The major volatile constituents of *Origanum majorana* are showed in the chromatogram are as follows: *trans*-Thuyanol 3.44%, terpinene 4-oil 23.6%, alpha-Terpineol 3.1%, *cis*-Tuyhanol 15.27%; alpha-Therpineil 3.1%, Sabinene 8.27%, and other components (less than 0.01%). Table 1 describes the physicochemical properties of *Origanum majorana* essential oiln (see Table 1 and Table 2 and Figure 1).

### 2.2. Study Groups

All procedures, including the informed consent process, were conducted in accordance with the ethical standards of the Helsinki Declaration of 1975, as revised in 2000. All subjects have been properly instructed and they consented to participate by signing the appropriate informed consent forms, being the ethical procedure approved by the Institution (#CIROM #014-2016). CIROM Center has been approved and certified by AENOR, Murcia, Spain (Spain; CIROM CERTIFICATE for dentist services; CD-2014-001 number; ER-0569/2014, following UNE-EN ISO 9001:2008 and UNE 179001-2001 Directive (Murcia, Spain). All patients followed the inclusion criteria described here. All efforts have been made to protect patient privacy and anonymity.

The average age of participants is 45 years old; all enrolled participants visit CIROM dental clinic for rutinary dental evaluation. They have a medium/high sociocultural status and 80% of them are living in Murcia (a metropolitan area in Spain) and 20% are from Alicante and Valencia (Spain, Europe). All participants completed the Hamilton scale II for anxiety/stress evaluation before/after NeurOptimal (NO) overtraining sessions (08:00 to 14:00). Patients are classified according to their Hamilton scores (scale II), including controls, since we also included controls (without stress) and NO trained participants during 21 training sessions (without bruxism).

All trained participants were naive to NeurOptimal neurotechnology procedures and had never participated in any neurofeedback study. NO sessions were conducted by an experienced and certified trainer from the Zengar Institute (Montreal, QC, Canada). Each NO learning takes 1 h/session (total: 21 h/participant) and all behavioral studies were concluded within 120 days (4 months). Finally, the Hamilton scale is completed by all enrolled participants, including controls and Bruxismitle participants with/without exposure to the essential oil during NO overtraining.

#### 2.2.1. Inclusion Criteria

We enrolled 89 participants, and 590 salivary samples were collected between 08:00 and 14:00 for cortisol evaluation by ELISA (ng/mL), following the manufacturer’s instructions. The study included these groups: (a) -Bruxismitle patients trained in NeurOptimal without smelling *Origanum majorana* essential oil during 21 training sessions (*n* = 12; 120 cortisol samples); (b) a control group. Each NO learning session takes 30 s at pre-learning (PRE) and 33 min and 30 s during post-training (POST) session. Each session training is repeated for 21 NO sessions. Additionally, a control group of participants without stress or bruxism were overtrained during 21 sessions with NeurOptimal (*n* = 5, 50 cortisol samples), including 30 controls (unstressed) without NO training (*n* = 30, 150 cortisol samples). In addition, 20 Bruxismitle participants were enrolled without NO training (*n* = 30, 150 cortisol samples). Since NO changes are expected to occur after six sessions by repeated experience, we decided to collect salivary samples after each six NO training sessions [session 1, (S-1), session 6 (S-6), session 12 (S-12), session 18 (S-18), and the last one, session 21 (S-21)], which were collected each six pre- and post-training sessions in all study groups; thus, the total number of salivary samples for cortisol quantification in Bruxismitle participants was 120 (*n* = 12, 120 cortisol samples)/without *Origanum majorana* odor exposure during NO overtraining (*n* = 12, 120 cortisol samples).

Brain activities were measured at PRE (pre-training), POST (post-training), being the parameter divergences, the difference of cerebral activity reached at NO pre-learning session minus values at post-training sessions (21 sessions) in Bruxismitle participants with/without *Origanum majorana essential* oil exposure during each session. The Hamilton scores also evaluated the degree of stress after NeurOptimal overtraining in Bruxismitle participants (*n* = 12); the aim is to evaluate whether NO overtraining could reduce cortisol levels. We compared their scores at pre-learning (PRE) and post-training values (POST, *n* = 30, see Table 2). These brain activities were compared among sessions, as well as compared to their initial NO session 1 (S-1) for Bruxismitle participants with/without *Origanum majorana* exposure during NO training (*n* = 12 patients * 21 sessions = 252 sessions, total: 504 data, see Table 3).

#### 2.2.2. Exclusion Criteria

We have excluded participants who had metabolic diseases/diabetes, brain disabilities, neurological/psychiatric disorders, liver/kidney disease/lupus/autoimmune disease/thyroid diseases/adrenal disease/Cushing syndrome/tumors. In addition, we also excluded women with menopause or patients taking regular medication (stimulants, anticonvulsants, antidepressant, or psychiatric/bipolar drugs), or those taking nutritional supplements, including chelators or alcohol intake.

We also excluded those participants who take caffeine or other stimulants, 24 h before collecting salivary samples.

### 2.3. Hamilton Stress Scale II in Patients

The Hamilton stress scale II is useful for self-stress and/or anxiety assessment in patients. All participants had completed the Hamilton survey (scale II) and their scores were compared at pre- and NeurOptimal post-training sessions with/without *Origanum majorana* essential oil exposure during each NO session. Bruxismitle participants had moderate scores (between 7 and 12), with 20 being the highest stress degree. Scores below 5 are indicative of absence of stress (controls) and values between 6 and 12 are representative of acute stress. Finally, scores higher than 13 (>13) indicate higher stress/anxiety level.

### 2.4. Cortisol

All participants were asked to provide salivary samples for cortisol assessment by providing salivary samples (08:00 to 12:00) since salivary levels correlate with plasma cortisol levels. The cortisol salivary levels were expressed as ng/mL by ELISA method, following the manufacturer’s instructions, and also avoid venipuncture. Salivary samples were collected during their current visit to CIROM clinic (between 08:00 and 14:00) for a rutinary dental evaluation. Cortisol levels remain stable in salivary for several days before follow-up [33,34,35,36]. Although they were stored at minus 70 °C. The analytical range of the assay is 0.024–30 ng/mL and the sensitivity for salivary cortisol detection is 0.024 ng/mL (Demeditec Diagnostic, Germany). Assays were conducted by interpolation within the standard curve following the manufacturer’s instructions (0 ng/mL to 30 ng/mL, as well as the range of standard curve between 0, 0.1, 0.4, 1.7, 7, and 30 ng/mL.

### 2.5. Materials and Methods

#### 2.5.1. How Does NeurOptimal Work

NeurOptimal^®^ (NO) monitors the brainwaves and gives subtle cues to the brain when it is not functioning smoothly. NeurOptimal^®^ (NO) is a non-invasive protocol (NeurOptimal^®^, NO, Zengar Institute, Montreal, QC, Canda)). Neurofeedback (NFB) indicates the way it is operating that the brain is using waves of cerebral activity (EEG analysis, electroencephalogram). The brain uses visual and auditory information offered by NeurOptimal^®^ to re-organize brain activity. The engineering principle of process control is optimized when the measurements of the data stream are accurate, the feedback to the process is both quick and precise, and the process adjustment is performed only when statically required to promote a directional jump shift or transformation to the next level of organization. No other neurofeedback systems have used the principle of live process control (Zengar Institute, Montreal, QC, Canada; https://neuroptimal.com/ accessed on 7 June 2020). When the brain is performing fluidly, NeurOptimal^®^ plays music; but if the brain’s activity begins to become inconsistent or less smooth, the music and images (fractals in the screen) are interrupted momentarily.

NeurOptimal^®^ 1.0 software dynamically controls patient feedback using nonlinear statistics to calculate the precise timming that feedback is given. The system responds to the patient’s brain wave information, and they respond to the system feedback. The interruption gently prompts the brain that it isn’t performing optimally. All the learning happens outside conscious awareness. As the brain begins to operate more efficiently, NeurOptimal^®^ adjusts itself automatically in response to the reached brain’s activity, individualizing the training microsecond-by-microsecond to improve the brain’s functioning. In this manner, learning continues based on the brain’s response to training. The participant must avoid moving large muscles or clenching their teeth since the electrical signal strength needed to move muscles will swamp the brainwave signals. The primary feedback is auditory, and the visual feedback is supportive during NeurOptimal training.

#### 2.5.2. What Is the Mean Divergence

NO registers the total electrical brain activity at pre-/post-training during each learning session. Brainwave frequencies are measured in hertz, defined as wave cycles per second. These brain frequencies range from the lowest and slowest Delta waves to Theta waves, to Alpha waves, to Beta waves, to the highest and fastest Gamma waves above 38 Hz. Bandwidths of brain frequencies are associated with different states of consciousness. NeurOptimal measures the total brain activities by divergence (DIV), which is an index of brain efficacy. Divergence measures the difference of cerebral activity reached at pre-training (PRE) minus those detected at post-training in each NO session. Divergence are positive or negative depending on if post-training values are higher or lower that pre-training values, respectively. The brain uses this information to reduce or increases its activity. Consequently, divergence (DIV) can reflect “autoplasticity” and cerebral activation. As the brain begins to operate more efficiently, NeurOptimal^®^ adjusts itself automatically in response to the brain’s activity, individualizing the training microsecond-by-microsecond in participants. Thus, NeurOptimal^®^ can adjust the necessary brain activity as a consequence of repeated NO training, leading to a lack of cerebral activity by overtraining.

##### Divergence a Measure of Stability

The brain activity at pre- and post-training at CAC graphs is a guide to evaluate the progress of NO training during sessions. Divergence reflects the relative stability of the brain. The lower the number the more stable the patient’ nervous system is. Progress is not linear, meaning that the divergence numbers do not go down in a straight-line orderly fashion. When divergences are higher it means that information has not been integrated at that session.

Thus, information is trying to be learned without success, and negative divergence suggests the information has not been integrated at that particular NO session yet. However, when brain activity post-training is close to pre-training levels, or even higher, the information has been integrated at that particular session (negative divergence or close to zero).

##### NeurOptimal^®^ Technology

How does NeurOptimal Client Hookup work? NeurOptimal^®^ Client Hookup consists of silver electrodes, applied to the clients’ ears and scalp, centered between the ear and the crown of the head on the bony ridge (central point C3 and C4). The electrodes are applied with EEG paste. It is water soluble electrical conductance material composed primarily of salts that enhance monitoring of the minute electrical pulses of the brain during pre-/post-training sessions (from 1 to 21 session).

##### Z-amp™ Amplifies Signals

The electrode sensors pick up the brain’s electrical signal and send that signal down a conductance wire to the Zengar Z-amp™ (NeurOptimal; Zengar Institute, Montreal, QC, Canada, https://neuroptimal.com/ accessed on 7 June 2020). This Z-amp™ cleans line noise and amplifies the brainwave signal. Other neurofeedback data looks “smeared” compared to NeurOptimal’s data due to a sampling rate of 256 samples per second, which are coupled with high precision of filtering, targeting, and triggering of feedback. At no time are electrical signals fed back to the brain.

The participant hookup consists of silver electrodes, applied to the patient’s ears and scalp, centered between the ear and the crown of the head on the bony ridge (central points C3 and C4). The electrodes are applied with EEG paste. It is water soluble electrical conductance material composed primarily of salts that enhances the monitoring of the minute electrical pulses of the brain (https://neuroptimal.com/ accessed on 7 June 2024).

##### Signal Separation into Frequencies and Intensities

The left and right brain wave signals are then separated by the computer software into their component frequencies and intensities.

Z-amp™ amplifies signals. The electrode sensors pick-up the brain’s electrical signal and send that signal down a conductance wire to the Zengar Z-amp™. The Z-amp™ filters line noise and amplifies the brainwave signal. Other neurofeedback data looks “smeared” in comparison NeurOptimal’s^®^ data, due to a sampling rate of 256 samples per second, coupled with high precision of filtering, targeting, and triggering of feedback. At no time are electrical signals fed back to the brain. Signal separation into frequencies and intensities left and right brainwave signals are separated by the software into their component frequencies and intensities. Intensity, for these purposes, is defined as a measure of the amount of electrical signal generated. This continuous data set is analyzed over time using non-linear statistics to determine when they enter into a period of “unstable” operation. Unstable periods are identified within milliseconds; feedback is given in the form of a pause in the music. Feedback is determined by continuously tracking the three variables of time, frequency, and intensity. This is called joint time frequency analysis (JTFA). The Zen modes of the NeurOptimal^®^ software automatically moves through four different modes called Zen 1, 2, 3, and 4. These modes are similar to a warmup stretch, a weight training challenge, a cardiovascular endurance, and a cool down period to integrate the learning (https://neuroptimal.com/ accessed on 7 June 2024).

##### Non-Linear Statistical Analysis of Data

This continuous data set is analyzed over time using non-linear dynamic math and statistics to determine when the brain enters into an area of “unstable” operation; feedback is given instantly within milliseconds. Feedback is given in the form of a pause in the music that is listened to and a momentary hesitation of the fractal image appears in the screen of the computer. This means that brain disturbances are associated to a pause in the music at pre- and post-learning sessions. These unstable periods are identified within milliseconds and feedback is given in the form of a pause in the music. Feedback is determined by continuously tracking the three variables of time, frequency, and intensity. This is called joint time frequency analysis (JTFA). The Zen modes of the NeurOptimal^®^ software automatically moves through four different modes called Zen 1, 2, 3, and 4. These modes are similar to a warmup stretch, a weight training challenge, a cardiovascular endurance, and a cool down period to integrate the learning (https://neuroptimal.com/ accessed on 7 June 2024).

##### Variables of Time, Frequency and Intensity

Feedback is determined by continuously tracking the three variables of time, frequency, and intensity. This is called joint time frequency analysis (JTFA).

##### Dynamic Dance

The continuous data set is analyzed by the software using 16 different target filters simultaneously. Each of these targets works dynamically with the brain at that exact time. The result of this “dance” between the participant’s brain and the NeurOptimal^®^ 1.0 software is broad-based and integrated in patient transformation.

##### Initial Learning Session

During the initial session, the expert NeurOptimal (MEC) trainer instructed the participants before NO performance. During the upcoming 21 learning sessions, the total brain frequencies have been measured at pre- and post-training sessions (total: 21 sessions/patient, approximately 1 h/patient). During each learning session, all participants must be comfortably reclining in a cozy chair with a warm blanket in a darkened room. The client has EEG sensors attached and a set of stereo earphones on. At the end of each session, the brain activity at pre-test session takes 30 s (pre-test; PRE, 15 s with open eyes and 15 s with closed eyes) and 33 min and 30 s during each post-training session. In addition, salivary samples were collected after pre-/post-training at certain sessions (session 1 (S-1), session 6 (S-6), session 12 (S-12), session 18 (S-18), and session 21 (S-21) for cortisol analysis. Immediately after concluding pre-training (PRE), salivary samples were collected for cortisol determination and, after concluding the post-training phase, we also collected salivary samples at certain sessions (S1, S6, S12, S18, and S21). The cerebral brain activity was measured after pre- and post-training in 21 sessions to monitor the efficacy of training in Bruxismitle participants. Salivary samples collection and NO performance was performed two times a week between 8:00 and 14:00, and NO training was concluded after 3 months.

##### Pre-Baseline Graphical Results and PRE/POST NeurOptimal Training

The pre-learning phase shows the baseline for cerebral activity. After each NO session, NO shows a spectrograph (Left) image and the CAC graph (Cross correlation of the auto correlation on right) after post-training. The software shows several fractals as a guide post-training for progress in total brain activities. The baselines are only a 30 s snapshot in time (PRE).

##### The Training Period

During each pre-learning session, music is listened to (30 s) (PRE) as and during 33 min and 30 s in the post-training phase (POST). Each participant will listen to NeurOptimal^®^ soothing instrumental new age music (the same piece for all participants). They will also be watching a monitor with a kaleidoscope of engaging fractal images. The optimal length of neurofeedback training series seems to be close to twenty, but the efficacy of NO training is expected to occur in six sessions. In our study, NeurOptimal training was performed for 21 sessions (2 times/week) to confirm the persistence and endurance of changes in Bruxismitle participants. Over NeurOptimal^®^ training is not a problem, if needed or desired, a person may elect to perform training every day. Usually there will be noticeable shifts within 6–10 sessions.

##### Patient Given Feedback

Graphical displays of brainwave variation appear during the training period, many different displays (spectra left and helix right) represent the minute-by-minute unfolding of the patient’s brainwave activity. The brain activity intensity will trigger the various frequency target filters as the training unfolds. When the nervous system statistically enters into an unstable operating area, feedback is given very rapidly, with precise timing, within milliseconds. Feedback is given in the form of a very brief pause in the music and a momentary hesitation of the fractal image, which means that some brain disturbances could occur exactly at that moment. Sometimes the pauses are so quick you may not consciously notice them. This pause is quite literally the pause that refreshes. A pattern of pauses are given to the brain. The brain immediately recognizes that these patterns are important and begins to self-reorganize. “When it comes to the frequency of feedback, LESS IS MORE” This means that divergences drops to zero (negative divergences). Consequently, brain activity is progressively decreased after post-training. These divergences identify when brain activity is increasing and its downregulation to identify brain reorganization in patients. A large drop in divergence numbers from before the training session to after indicates that the brain is working hard to shift into regulation to finally reach a calm state (divergences close to zero or even negative). The NeurOptimal^®^ 1.0 software has four different modes called Zen 1, 2, 3, 4, 5 (see below, i.e., Zen 5 version mode of 45 min). NeurOptimal training was performed for 33 min and 30 s by session in our study.

##### Post-Baseline Graphical Results

The trainer will compare the pre- and post-spectrograph (Left) and the pre- and post-CCAC graph (cross-correlation of the auto-correlation on right) found in each NO session (total 21 sessions). The pre-test (PRE) session will show a total frequency as well as other differences at post-training in each session (see Table 4 and Table 5).

### 2.6. Statistical Analysis

The statistical analyses were evaluated by SPSS (V17.0), and Sigma Plot software (11.0). Mean and standard deviation were estimated at pre- and post-training in all 21 training sessions [from initial session 1 (S-1) to the final session 21 (S-21)]. To investigate the effect of NeurOptimal training on Divergences, we performed ANOVA for repeated measures. The Greenhouse-Geisser indicates the significant effect (*p* < 0.05) or not (*p* > 0.05) following the repeated ANOVA analysis and W Machutly has been used to detect sphericity.

The main hypothesis was examined using repeated analyses of variance, including divergences as the repeated-measures factor, group as the between-groups factor (bruxism with/without *Origanum majorana* stimulation), and cortisol levels at certain sessions (S-1, S-6, S-12, S-18, S-21) as dependent measures. Data were expressed as mean+-relative error. A One-Way Kruskal–Wallis test investigated the effect of training on brain activity at pre- and post-training, separately for pre- and post-training. The interactive effects of the essential oil *Origanum majorana* stimulation and brain activities (divergence, DIV) were also evaluated by bifactorial ANOVA analysis. In this case, we have considered the stress factor at NO post-training, the aromatherapy factor (*Origanum majorana* odor), as well as the interactive effect at post-training (ST)*Aromatherapy factor (*Origanum majorana* fragrance fragrance).

A Mann–Whitney test evaluated comparisons between nonparametric data without homogeneity of variance when a Levene test is significant (*p* < 0.05). However, we have analyzed data by a Bonferroni test following when the Levene test is not statistically significant (*p* > 0.05), reflecting that data following a normal distribution (homogeneity of variance). A two-sided *p* value of <0.05 was statistically considered if *p* < 0.05, and highly statistically significant when *p* < 0.01. We have studied the effect of training on cortisol samples at certain sessions (S1, S6, S12, S18. S21) during pre- (PRE) and post-training (POST) by a measured repeated ANOVA analysis. Kruskal–Wallis analyses have shown certain significant changes on cortisol or brain activities (pre-, post-training, or DIV) in Bruxismitle participants during 21 sessions, separately. In addition, the Spearman test evaluates significant correlation between cortisol samples and divergences (*p* < 0.05). The divergence is the difference of cerebral activity in Bruxismitle participants stimulated with *Origanum majorana* minus brain activities of Bruxismitle participants without *Origanum majorana.*

## 3. Results

The repeated ANOVA indicated a lack of significant effect for total divergence (index of cerebral activity), including positive and negative values [F (1,20) =1.27; *p* = 0.2; n.s, Etha square = 0.14, power = 0.85) according to Greenhouse–Geisser data (Figure 2).

Figure 2 shows divergences between participants. Divergence is the difference of reached cerebral activity found at pre-learning (PRE) minus post-training (POST) values with/without *Origanum majorana* during NO training. Brain activities were assessed in 12 Bruxismitle participants without *Origanum majorana* stimulation (252 measurements of cerebral activity), as well as in 12 Bruxismitle stimulated participants with this fragrance (252; total: 504).

Figure 3 shows representative divergences in Bruxismitle participants with elevated brain activities at post-training compared to pre-training values with different degrees.

### 3.1. Effects of NeurOptimal Pre-Training on Brain Activities (Divergence) in 21 in Bruxismitle Patients with/Without Origanum Majorana Stimulation

The Kruskal–Wallis analysis indicated a statistically significant increase in cerebral activity at pre-training (PRE) after 21 training sessions [(H (1,20) = 44.35; *p* < 0.001)]. The post hoc analysis at pre-training confirmed moderate raises in Bruxismitle patients after *Origanum majorana* inhalation, exactly at certain sessions (S-2, *p* < 0.05), session 11 (S-11, *p* < 0.05), session 13 (S-13, *p* < 0.05), session 14 (S-14, *p* < 0.05), including the last session (S-21, *p* < 0.05); all of them compared with their initial values at session 1 (S-1; Figure 4).

### 3.2. Effects of NeurOptimal Post-Training on Brain Activities in Bruxismitle Patients Origanum Majorana(Without) Stimulation

The Kruskal–Wallis analysis showed that NeurOptimal post-training (POST) significantly increases brain activities in Bruxismitle patients after 21 sessions without *Origanum majorana* stimulation [H (1,20) = 28; *p* < 0.05]. In fact, Mann–Whitney confirmed a rise in cerebral activities at certain post-learning sessions (sessions S12, S16, S18 and S21, all compared with the session 1 (*p* < 0.05, S-1; Figure 5)

The Kruskal–Wallis analysis also showed that NO post-training trend to increase brain activities by *Origanum majorana* stimulation in Bruxismitle participants [(H (1,20) = 28; *p* = 0.09; n.s)]. Thus, *Origanum majorana* inhalation during NO post-training could lead to a moderate rise in brain activity as compared to the initial learning session (S-1; Figure 4). In fact, Mann–Whitney confirmed these higher brain activities at certain post-training sessions after *Origanum majorana* inhalation in Bruxismitle patients (session S-6, S-12, S-18, S-20, and S-21) as compared to the session 1 (S-1, Figure 4 and Figure 5).

Figure 5 shows representative divergences in participants and brain activities at pre-/post-training (PRE and POST) in Bruxismitle participants with different post-training as compared to pre-learning sessions (Figure 6) or those close at pre- and post-training sessions (Figure 6).

This Figure 7 indicates NeurOptimal performmance examples of low and elevated brain activities. Divergences are shown in white color (DIV) while orange color (PRE) shows cerebral activity at pre-learning session as compared to yellow color (POST), which is the cerebral activity at post-training session (Figure 7).

The following image indicates several values of cerebral activities at PRE and POST, as well as divergences during 21 NeurOptimal sessions (NO (Figure 7) in Bruxismitle participants with/without *Origanum majorana* odor stimulation (Figure 7, Figure 8 and Figure 9).

### 3.3. Analysis of Negative Divergences for Brain Activity Decreased in Bruxismitle Participants After 21 NeurOptimal Training Session with Origanum Majorana Essential Oil Inhalation

Figure 10 shows mean cerebral activities in Bruxismitle participants without/with the essential oil (21 × 12: 252 measure of PRE data and 252 measures of post-training data (POST). The repeated ANOVA failed to detect a significant effect for divergences [F (1,20) = 1.27; *p* = 0.2; n.s, Etha square = 0.14, power: 0.85) according to Greenhouse–Geisser data (Figure 10).

The Kruskal–Wallis analysis failed to detect a significant reduction in brain activities by overtraining in 21 sessions [H (1,20) = 22.54; *p* = 0.32, n.s]. However, these divergences were progressively lower in Bruxismitle participants after *Origanum majorana* exposure during NO overtraining (Figure 10).

The Mann–Whitney analysis indicates that *Origanum majorana* odor exposure decreased brain activities in Bruxismitle participants as compared to their initial session 1 (S-1, *p* < 0.05), leading to a final calm state by reducing divergence, which trends to zero as a consequence of repeated NO training (308 + −86 vs. 785 + −181 value for divergence; *p* < 0.05, Figure 10 and Figure 11). At session 16 (S-16), *Origanum majorana* exposure in Bruxismitle participants significantly decreased brain activities in comparison to Bruxismitle participants without *Origanum majorana* odor stimulation during NO overtraining (*p* < 0.05). However, *Origanum majorana* odor exposure during NO overtraining significantly increased total brain activity at session 8, reflecting fluctuations on electrical activity associated to high cerebral activities by anxiogenic effects or unpleasant odors (*p* < 0.05). Thus, a decrease in brain activities was observed at session 16, which means a calm state is almost reached after concluding 16 NO sessions.

### 3.4. Hamilton Scale II Scores Are Reduced at Post-Training Session in Bruxismitle Patients as Compared to Their Pre-Learning Session

Patients with bruxism with moderate stress according to their Hamilton scores (score: 18) were compared with healthy control subjects without stress (score: 5, *p* < 0.05). Their scores before NeurOptimal training are 15 while reaching a value of 12 after NeurOptimal overtraining (21 sessions, *p* < 0.05, Figure 12). Thus, NeurOptimal post-training promotes a 44% reduction of stress degree after 21 post-training sessions as compared to their Hamilton scores at the pre-learning session (*n* = 30, score: 18).

### 3.5. Effects of NeurOptimal Training on Salivary Cortisol Levels (ng/mL)

Cortisol levels were significantly reduced by NeurOptimal overtraining in Bruxismitle participants expose to *Origanum majorana* odor compared to Bruxismitle participants without NO training (*p* < 0.05); thus, NO training could provide resilience against stress-related conditions in bruxism.

#### Cortisol Levels (Salivary, ng/mL)

Cortisol levels were measured in 89 participants (590 salivary samples) by ELISA. The repeated ANOVA showed a significant effect of NeurOptimal training on salivary cortisol levels at sessions 1, 6, 12, 18, and 21 (S-1, S-6, S-12, S-18, and S-21) by NO overtraining [F (1,4) = 3.6; *p* < 0.05, Etha square: 0.14, powder: 0.85)]; there was a trend for an interaction between brain activities (divergences) and salivary cortisol levels [F (1,4) = 1.8; *p* = 0.1, Etha square: 0.07, power: 0.53]. The effect of *Origanum majorana* exposure during NO training and cortisol interaction is also statistically significant [F (1,4) = 2.98; *p* < 0.05, Etha square: 0.11, power: 0.77]. 

The bifactorial ANOVA allows us to evaluate the effect on 378 salivary samples by testing the factor Bruxismitle at NO post-training, the effect of *Origanum majorana* exposure factor or possible interactive effect among both factors (Bruxismitle and *Origanum majorana* exposure during each NeurOptimal session). With regard to cortisol levels, the bifactorial ANOVA confirmed decreased cortisol levels in Bruxismitle by NeurOptimal overtraining [POST, F (1374) = 4.5, *p* = 0.035, *p* < 0.05]. The observed synergic anxiolytic effect associated to NeurOptimal overtraining in Bruxismitle participants exposed to this odor was also evident [F (1374) = 4.5, *p* = 0.035)]. However, *Origanum majorana* factor by itself did not induce anxiolytic effects in control participants [F (1374) = 2.61; *p* = 0.1, n.s)]. Finally, a trend to reduce cortisol levels in Bruxismitle patients exposed to *Origanum majorana* was confirmed as compared to controls (without bruxism, *p* = 0.1, n.s).

The cortisol levels are significantly reduced in Bruxismitle participants at post-training (POST) when they are exposed to *Origanum majorana* odor during NO overtraining as compared to pre-learning sessions (PRE, *p* < 0.05). Cortisol levels were significantly reduced in Bruxismitle participants by repeated NO training (POST) at session 12 (S-12) as compared to their pre-learning (PRE) values (*p* < 0.05). At pre-learning session 12 (S-12), Bruxismitle participants exposed to *Origanum majorana* odor had significantly decreased cortisol levels (Pre-AE) as compared their pre-learning values of participants without smelling this odor (PRE, *p* < 0.05), suggesting that NO overtraining could increase the effectivity of NeurOptimal during further sessions; in fact, at post-training session 12, Bruxismitle participants exposed to *Origanum majorana* (Post-AE) essential oil decreased their cortisol levels as compared their pre-learning data in those without smelling *Origanum majorana* odor during NO training (Pre-AE, *p* = 0.049). At NeurOptimal post-training sessions, Bruxismitle NO trained participants exposed to *Origanum majorana* odor (Post-AE) reduced their cortisol levels at session 1 (S-1) as compared to their own pre-learning (Pre-AE) values (*p* < 0.05). At session 1 (S-1), Bruxismitle participants exposed to *Origanum majorana* essential oil during NO post-training (Post-AE) had significantly decreased cortisol levels as compared to Bruxismitle participants without smelling this odor during post-training session Post; *p* < 0.05; see Figure 13 and Figure 14, and Table 6).

Additionally, there was a negative correlation between cortisol levels and divergences in Bruxismitle patients exposed to *Origanum majorana* odor during NO overtraining (r = −0.44, *p* = 0.05; *n* = 12).

Figure 15 indicates the percentage of salivary cortisol levels vs. controls, which are 100%, we confirmed a cortisol decrement after 21 NO post-training sessions in Bruxismitle patients as compared to controls (*p* < 0.05) without effect for other comparisons (*p* > 0.05, n.s in all cases).

## 4. Discussion

Neurofeedback is an effective way to train electrophysiological activity of the targeted cortical area in patients [34,35,36]. This is the first study that shows anxiolytic effects in Bruxismitle participants exposed to *Origanum majorana* by 21 NeurOptimal overtraining sessions. These anxiolytic effects by habituation in Bruxismitle participants agree with reported anxiolytic effects in patients suffering PTSD after neurofeedback training (post-traumatic stress disorder) [37], a neuropsychiatric condition associated with strong stress.

It is known that cortisol levels follow an inverted-U-learning curve in learning and memory processes [37,38,39,40,41,42] and accelerate the progression of certain psychiatric diseases. Our findings also confirm NO is a safe neurotechnology for preventing cortisol over-release in Bruxismitle participants. An EEG (electroencephalogram) is a recording of the electrical activity of the brain and these brainwave frequencies are measured in hertz (wave cycles per second). Brain frequencies range from the lowest and slowest Delta waves to Theta waves, to Alpha waves, to Beta waves, and to the highest and fastest gamma waves above 38 Hz. In our study, Bruxismitle participants experienced fluctuations in cerebral activity by increasing divergence at certain desired electrocortical activity associated to unpleasant odors (see session 16) but suppressing undesirable activity as a consequence of habituation (session 8) by NO overtraining [31,43]. *Origanum majorana* exposure during NO sessions increased brain activity at session 8 (S-8), perhaps suggesting undesired electro-cortical activity (session 8) to discard unpleasant odors at that particular session. Bandwidths of brain frequencies are associated with different states of consciousness. Thus, NeurOptimal neurotechnology working at 21 Hz (Aura frequency) is associated with an activating frequency, as occurred in session 8 (S-8), since the calm state is not reached at that session; however, this calm state is reached by session 16, which is associated with 38 Hz of frequency, while the frequency of 15–18 hertz is associated with feelings of safety (https://neuroptimal.com/ accessed on 7 June 2024). On the other hand, the lateralization of brain activation plays a role in the brain’s response to pleasant odors (right hemisphere) while unpleasant smells are registered in the left hemisphere [44]. In our study, all participants had cortical electrodes on both hemispheres.

*Origanum majorana* essential oil enhances anxiolytic effects by NeurOptimal overtraining in Bruxismitle participants.

The volatile constituent of *Origanum majorana* essential oil may enter the bloodstream and promote anxiolytic effects in Bruxismitle patients, leading to cortisol reduction; this effect is consistent with demonstrated anxiolytic effects in healthy subjects exposed to lavender oil fragrance [45,46]. In fact, these odors (fragrance or essential oils) can influence the reaction time in humans exposed to pleasant and unpleasant odors in the environment [8], which could explain these discrepancies on their cerebral activities found at 8 or 16 NO sessions. Since monoterpene derived from *Rosmarinus officinalis* oil can modulate the sympathetic nervous system activation [47], we should not exclude the possibility that certain terpenoids from *Origanum majorana* essential oil may induce anxiolytic effects in Bruxismitle participants exposed to this odor during NO training; in fact, essential oils can diffuse to the olfactory bulb and pass to the limbic system. These observed anxiolytic effects by NO overtraining could occur through limbic projections of the olfactory pathways [48] because cognitive improvements have been described after pleasant odors (peppermint) in subjects exposed to these odors. The constituents of *Origanum majorana* essential oil may induce anxiolytic effects and/or modulate brain activities by habituation as a consequence of NO overtraining in Bruxismitle patients in our study (Figure 2). However, *Origanum majorana* essential oil provokes a temporal unpleasant effect at certain NO sessions, as we observed in the session 8. In fact, elevations in brain activities (DIV) observed at session 8, reflected some specific unpleasant odor exposure without toxicity (Figure 3), which agreed with unpleasant effects in healthy subjects exposed to isovaleric acid from *Rosmarinus officinalis* odor [9,49,50]. The olfactory system mediates both learned and innate responses to odor stimuli. From this perspective, these observed fluctuations on cerebral activities in Bruxismitle patients by increasing their divergences at session 8 suggests they did not reach a final calm state; conversely, the habituation associated to NO overtraining lead to a calm state S-16, by reducing cerebral activities at this session (see Figure 4).

On the other hand, these positive peaks of divergence (cerebral activities) could be associated with discarded unpleasant effects at certain NO sessions (i.e., session 8). However, when divergences (DIV) at post-training times are negative or close to pre-learning values, the patients will reach a final calm state by habituation after 21 overtraining NO sessions. Consequently, patients reach a better brain reorganization after concluding 21 NO overtraining sessions. In fact, when divergences (DIV) become more negative, the brain is quickly and progressively reorganized and reaches a relaxation state by decreasing salivary cortisol levels by NO overtraining during 21 sessions. These mediated-anxiolytic effects of *Origanum majorana* exposure during NO training in Bruxismitle patients is supported by similar results in healthy subjects exposed to lavender and *Rosmarinus officinalis* essential oil. However, in these studies [9,50], odors are not directly impregnated in the nasal filters, as in our study; *Origanum majorana* odor constituents can reduce anxiety by reducing the stress degree in Bruxismitle participants in our study [51]. Lehmer et al. observed in healthy subject who smell lavender or orange essential oil fragrance experienced less stress during their visit to a dental clinic as compared to controls without smelling this odor [52]. This indirect evidence is consistent with the anxiolytic effects by NO overtraining observed in expose Bruxismitle patients to *Origanum majorana* odor. Moreover, the exposure to lavender essential oil in sheep during its transport was able to reduce cortisol levels [53]. Although lavender essential oil can induce autonomic deactivation of the sympathetic system by decreasing cAMP activity at the post-synaptic level [54], we cannot elucidate the exact molecular mechanism by which *Origanum majorana* oil decreased cortisol levels in our study. However, NO overtraining after 21 sessions by habituation lead to anxiolytic effects and reduced salivary cortisol levels. Perhaps olfactory stimulation modulates emotional changes by its direct impregnation on nasal filters in our study. Interestingly, cortisol changes at NO post-training sessions correlate with divergence in Bruxismitle patients in our study. Since neuroimage studies have confirmed that the olfactory system plays a role in modulating anxiolytic responses [55,56,57], the mechanism by which *Origanum majorana* has anxiolytic effects are as yet unknown. The cortisol reduction by *Origanum majorana* in Bruxismitle participants agrees with anxiolytic reported effects in healthy subjects exposed to essential oils as lavender oil [52], which also improved transthoracic Doppler echocardiography [58,59,60].

The parameter divergence is the difference in cerebral activity found at post-training sessions minus their pre-learning value during each NO session. Despite cortisol levels decreased by overtraining in our study, the individual variability among Bruxismitle participants provoked a lack of significant effect for divergence (Figure 3). In addition, these cortisol raises are not attributable to obesity or overweight in our participants because BMI (body mass index) did not change in our study (*p* > 0.05, n.s). Although we cannot elucidate the brain areas involved in the control of cerebral activity by NO overtraining, it is known that patients experienced better brain connectivity in the dorsomedial prefrontal cortex and amygdala after neurofeedback training [11,61]. After training, participants successfully self-regulated the top-down connectivity within these brain areas. Our findings confirm the safety of NO neurotechnology and its capacity to promote anxiolytic effect by decreasing salivary cortisol levels; this NO neurotechnology could be an alternative approach against benzodiazepines abuse in anxiety-related disorders. Benzodiazepines can suppress neural activity by regulating GABA-receptor activation in brain areas involved in the control of anxiety [62]. Given the concerns about benzodiazepines treatment, *Origanum majorana* essential oil emerges as alternative against stress related behaviors in Bruxismitle patients. In addition, *Origanum majorana* essential oil and the enhanced anxiolytic efficacy of NeurOptimal overtraining could decrease salivary cortisol levels in participants suffering bruxism.

## 5. Conclusions

The NeurOptimal (“a version of Neurofeedback”) system is a safe neurotechnology able to reduce stress-related behaviors. *Origanum majorana* essential oil decreased salivary cortisol levels in Bruxismitle participants with NO overtraining. In fact, neurofeedback technology (including NO) could induce functional changes in interconnected brain areas involved in the regulation of stress and anxiety response in patients. *Origanum majorana* significantly reduced cortisol levels in bruxism and may also modulate brain activities with repeated NO overtraining. Elevated cortisol levels contribute to the progression of certain neuropsychiatric diseases (i.e., PSTD/Anxiety). On the other hand, *Origanum majorana* exposure during NO training leads to fluctuations in brain activities (divergences), which could help to induce desired changes in electro-cortical activity and also suppress undesirable activity associated with unpleasant odors at certain NO sessions. In fact, *Origanum majorana* fragrance increased brain activity at session 8 (S-8), before reaching a final calm state by repeated NO overtraining. Collectively, our findings suggest NO neurotechnology is safe for preventing stress in Bruxismitle participants. In this way, stress contributes to the progression of neuropsychiatric disorders and neurofeedback technology can help to prevent certain CNS pathologies, such as depression [28,63] epilepsy [64], autism, attention deficit/hyperactivity disorder, etc. [65,66,67,68]. Thus, *Origanum majorana* essential oil decreased cortisol in Bruxismitle participants and may modulate alertness states. Our findings are consistent with some demonstrated beneficial effects of aromatherapy by modulating records of EEG activities in https://neuroptimal.com/ accessed 7 on June 2020 patients. A systematic review has confirmed the good efficacy of cortisol levels to identify bruxism in subjects. The possible relationship between bruxism and elevated salivary cortisol levels can be associated with the etiology of anxiety, fear, frustration, stress, and also bruxism [38]. In fact, the association between high salivary cortisol levels upon waking and bruxism has been confirmed in several studies [69]. The possible contribution of emotionally altered responses should not be excluded in bruxism [70,71,72,73,74]. Based on the available evidence, some meta-analysis suggests higher salivary cortisol levels is a biomarker of bruxism [75,76], in agreement with our findings. Finally, a meta-analysis confirmed the correlations between cortisol levels, bruxism, and the etiopathogenesis of temporal mandibular disorders by evaluating studies from 2008 to 2020 [27] and other published studies [74,75].

## Figures and Tables

**Figure 1 biomimetics-09-00715-f001:**
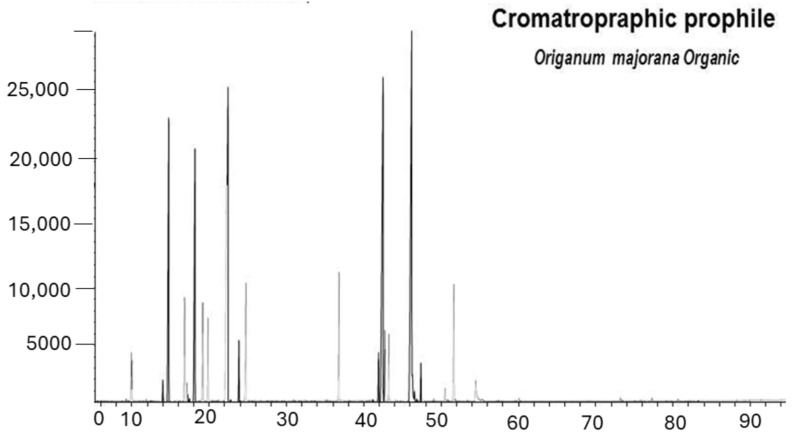
Chromatographic profile.

**Figure 2 biomimetics-09-00715-f002:**
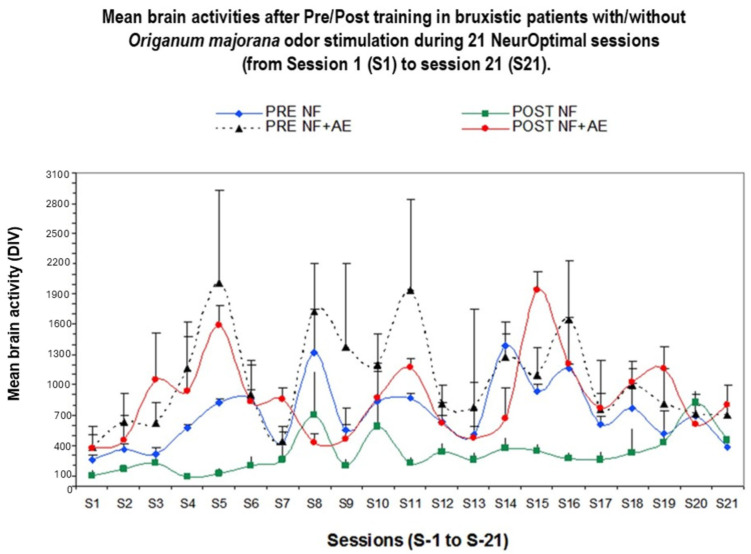
Effect of NeurOptimal training on cerebral activities at pre-learning and post-training during 21 sessions in Bruxismitle participants with *Origanum majorana* stimulation (*n* = 12, 252 measurements) as well as in Bruxismitle participants without exposure to this essential oil (*n* = 12, 252 measurements). **PRE**: Brain activities at pre-learning sessions in Bruxismitle patients (blue line). **POST**: Brain activities at post-training sessions in Bruxismitle patients (green line). **PRE-AE**: Brain activities in Bruxismitle participants at pre-learning exposed to *Origanum majorana* odor during NO training (black line). **POST-AE**: Brain activities in Bruxismitle participants exposed to *Origanum majorana* odor at post-training (red line).

**Figure 3 biomimetics-09-00715-f003:**
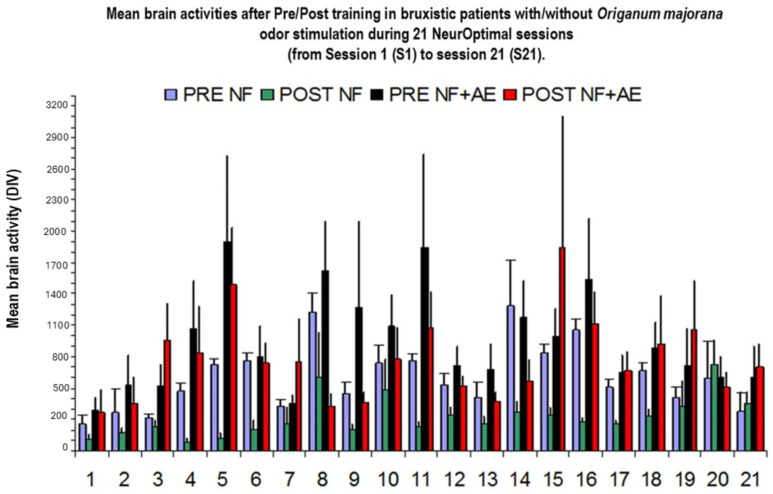
Divergences in all participants. **FPRE**: Brain activities in Bruxismitle patients at pre-learning sessions (blue column, Figure 3). **POST**: Brain activities in Bruxismitle patients at post-training sessions (green column, Figure 3). **PRE NF + AE**: Brain activities in Bruxismitle participants exposed to *Origanum majorana* odor at pre-learning sessions (black column, Figure 3). **POSTNF + AE**: Brain activities in Bruxismitle participants exposed to *Origanum majorana* odor at post-training in (red column, Figure 3).

**Figure 4 biomimetics-09-00715-f004:**
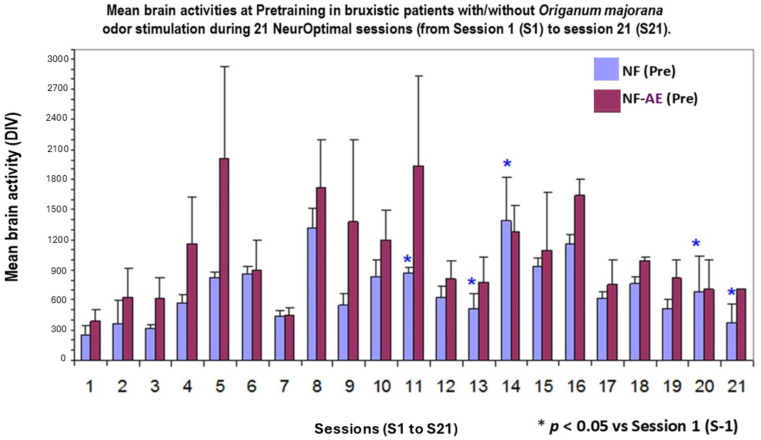
Mean cerebral activities at NeurOptimal training (PRE-TEST) in Bruxismitle patients with/without *Origanum majorana* odor exposure. **PRE-NF** (blue column): brain activities in Bruxismitle patients at 21 pre-learning sessions (PRE) without *Origanum majorana* inhalation. **PRE-NF AE**: Brain activities at 21 pre-learning (PRE) sessions in Bruxismitle patients with *Origanum majorana* exposure (red column, Figure 4). * *p* < 0.05 vs. session 1.

**Figure 5 biomimetics-09-00715-f005:**
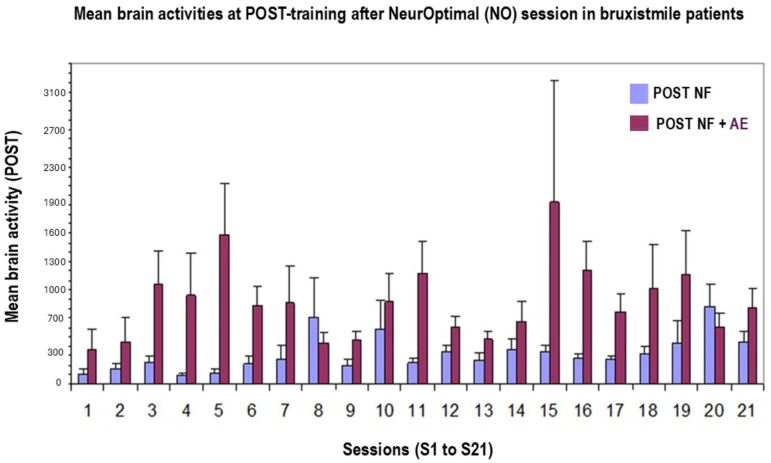
Divergences in all participants: brain activities after POST training without/with Origanum majorana odor stimulation during 21 NeurOptimal training sessions. **POST-NF** (blue column): Brain activities during 21 neurofeedback (NeurOptimal) sessions (PRE) in Bruxismitle patients without *Origanum majorana* exposure during 21 NO sessions. **POST-NF AE:** Brain activities at 21 pre-learning (PRE) sessions in Bruxismitle patients without *Origanum majorana* exposure during each NeurOptimal session (red column, Figure 4, *p* < 0.05 vs. session 1).

**Figure 6 biomimetics-09-00715-f006:**
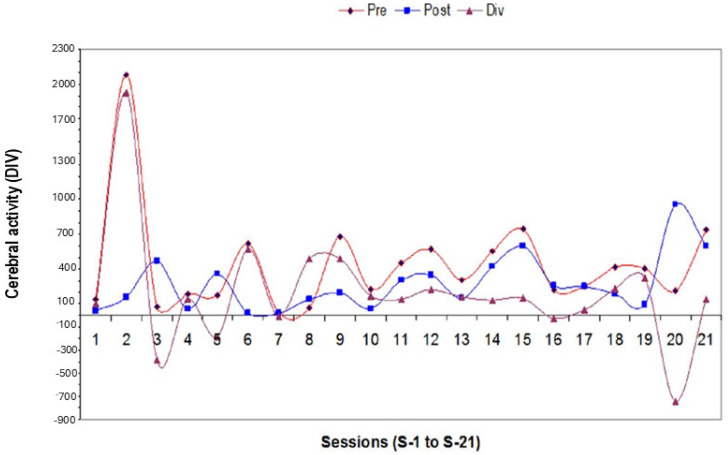
Example of divergences in a patient: representative brain activities at pre-/post-training (PRE and POST) in Bruxismitle participant.

**Figure 7 biomimetics-09-00715-f007:**
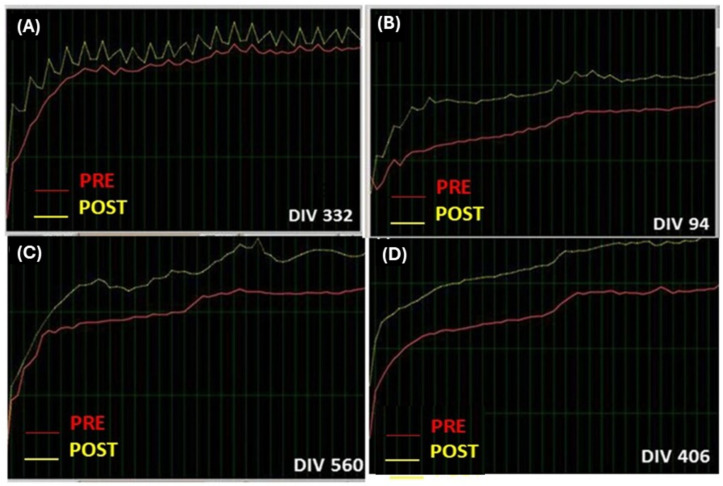
Examples of low, moderate, and high divergences (DIV).

**Figure 8 biomimetics-09-00715-f008:**
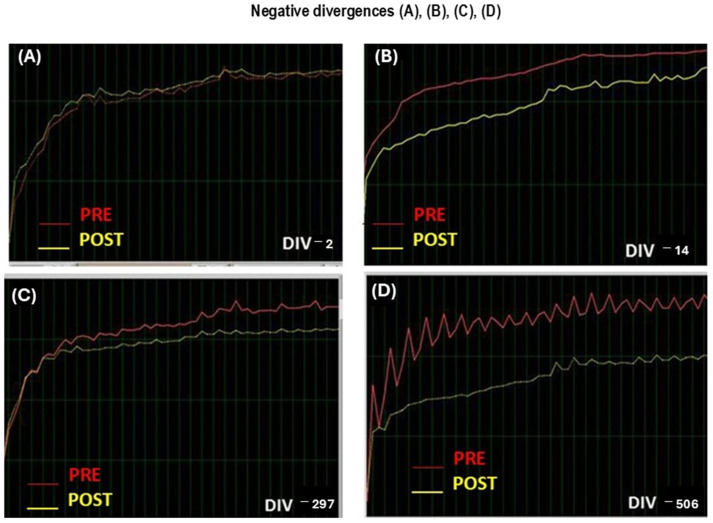
Examples of low, moderate, and high divergences (DIV).

**Figure 9 biomimetics-09-00715-f009:**
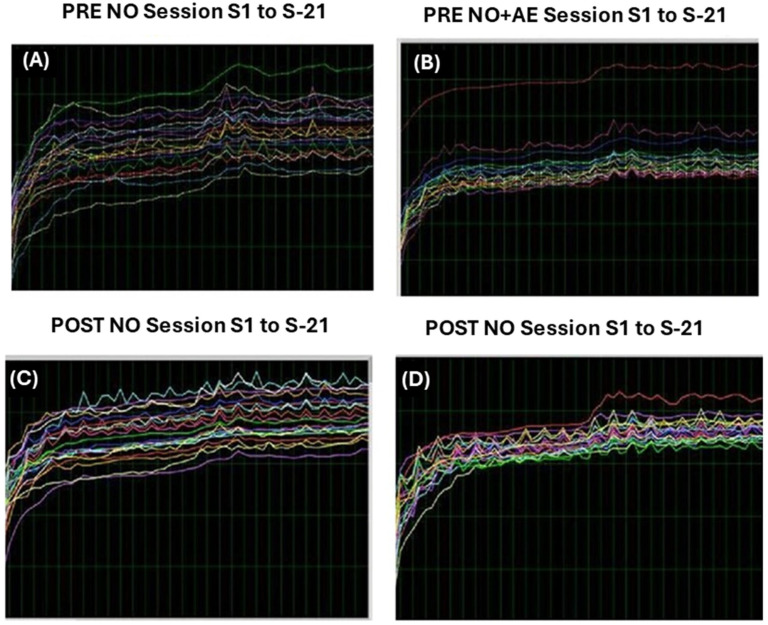
Several brain activities (DIV) in Bruxismitle participant at pre- and post-training with/without *Origanum majorana* exposure during NO session. These color lines are representative of each NO session (total 21).

**Figure 10 biomimetics-09-00715-f010:**
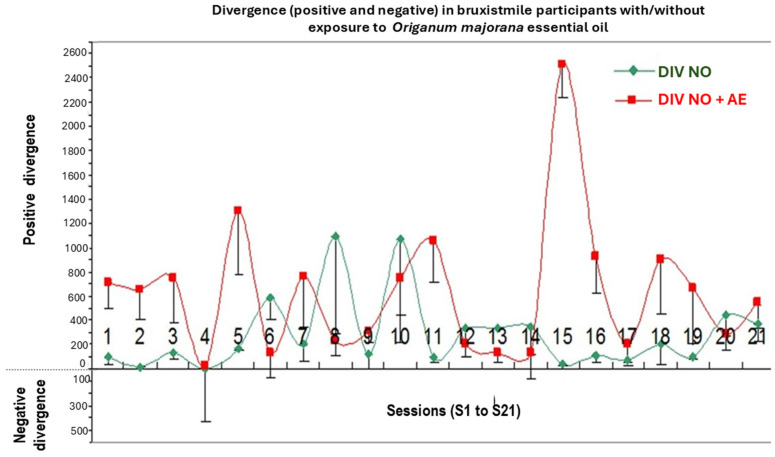
Negative divergence (DIV) in NeurOptimal training with/without *Origanum majorana* odor exposure during 21 NO sessions. **DIV Total NF: ** brain activities (total divergence—positive and negative—in Bruxismitle patients after 21 NeurOptimal sessions exposed to *Origanum majorana* odor in each session (green line, Figure 10). **DIV negative NF**: negative brain activities in Bruxismitle participants with *Origanum majorana* odor exposure during 21 sessions (red line, Figure 10).

**Figure 11 biomimetics-09-00715-f011:**
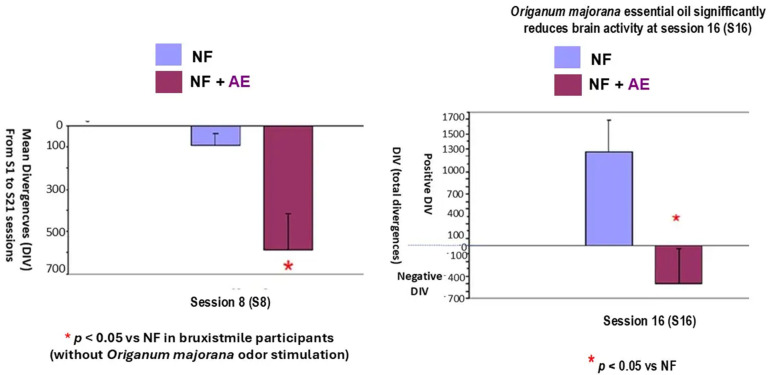
*Origanum majorana* odor reduces cerebral activities by repeated NeurOptimal overtraining in Bruxismitle participants as compared to Bruxismitle participants without exposure to this odor during each NO session. **NF: ** Total brain divergences (positive and negative) were evaluated in Bruxismitle patients after 21 NeurOptimal sessions (blue color) with a significant decrements on cerebral activity at session 8 (S-8, see left panel). **NF + AE:** Brain divergences (total Divergence—positive and negative) in Bruxismitle patients expose to *Origanum majorana* essential oil exposure session 16 (S-16).

**Figure 12 biomimetics-09-00715-f012:**
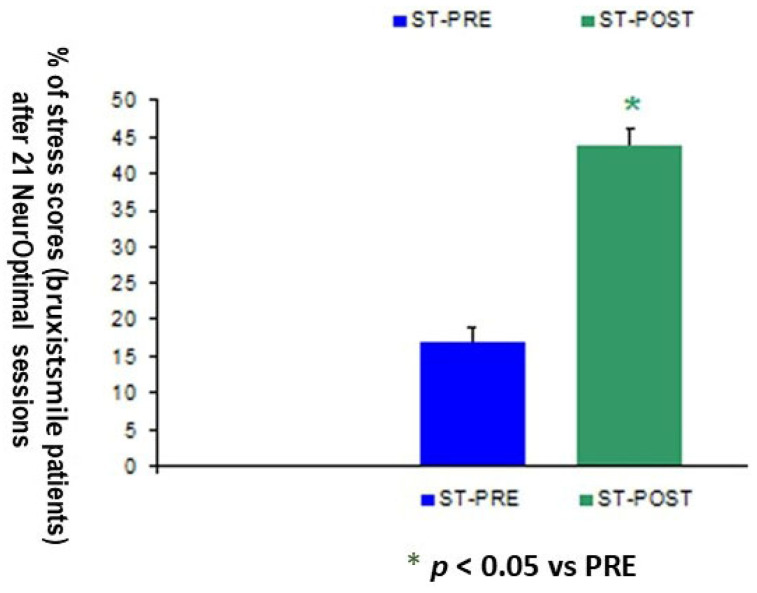
Reduced Hamilton scores after 21 NO post-training (POST) sessions in Bruxismitle patients as compare their pre-learning sessions (PRE).

**Figure 13 biomimetics-09-00715-f013:**
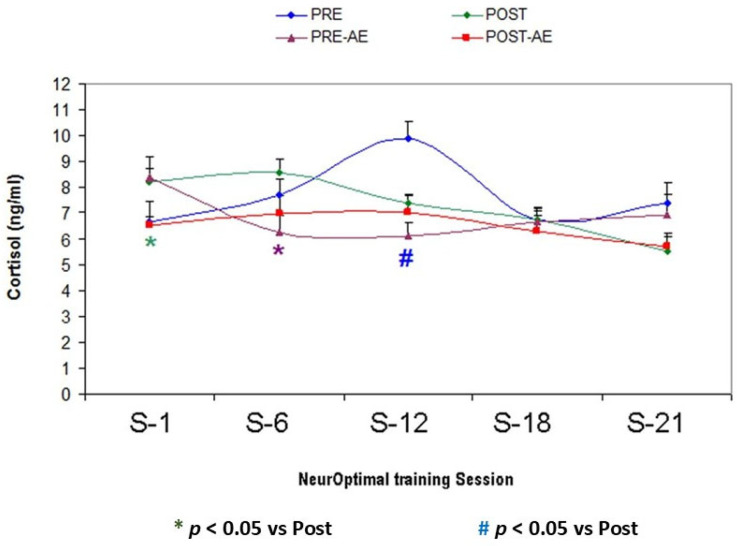
Anxiolytic effects of *Origanum majorana* essential oil in Bruxismitle participants after 21 NeurOptimal sessions by decreasing salivary cortisol levels.

**Figure 14 biomimetics-09-00715-f014:**
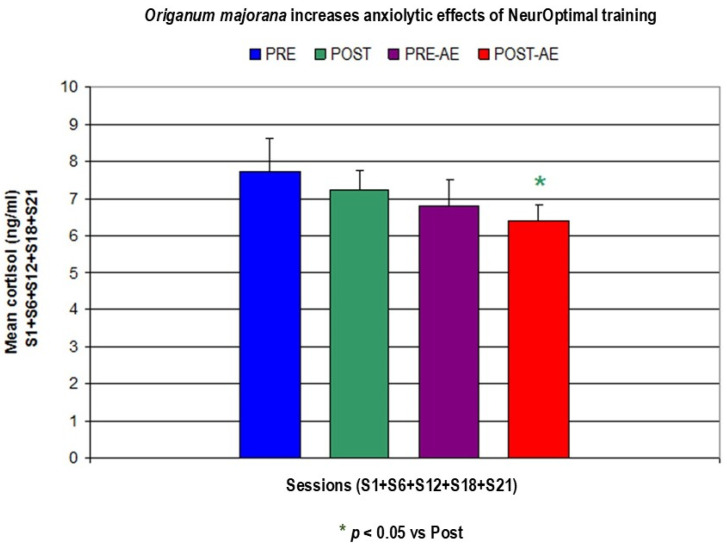
Mean salivary cortisol levels (ng/mL) in Bruxismitle participants with/without *Origanum majorana* exposure during 21 NO sessions (S-1 + S-6 + S-21 + S-16 + S-21).

**Figure 15 biomimetics-09-00715-f015:**
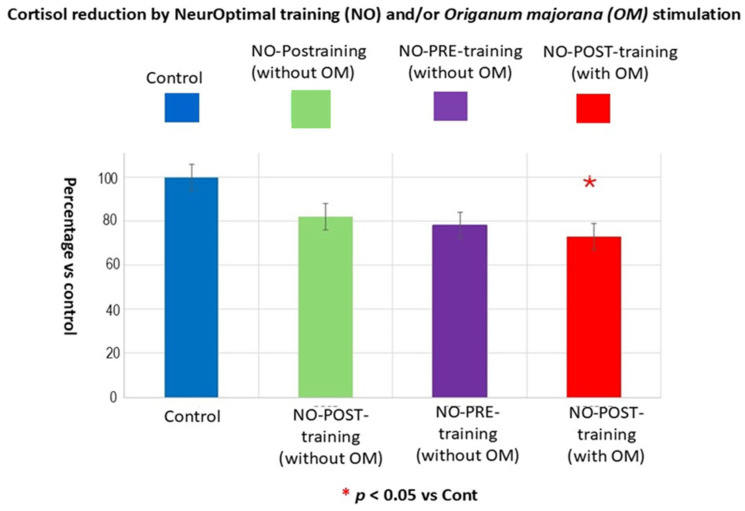
Percentage of cortisol levels vs controls. **PRE**: Brain activities in Bruxismitle patients at pre-learning sessions (blue column). **POST**: Brain activities in Bruxismitle patients at post-training sessions (green column). **PRE-AE**: Brain activities in Bruxismitle participants during 21 NeurOptimal sessions at pre-learning session with *Origanum majorana* essential oil (purpura column, Figure 15). **POST-AE**: Brain activities in Bruxismitle participants exposed to *Origanum majorana* odor at post-training (red column).

**Table 1 biomimetics-09-00715-t001:** Chromatographic chemical characteristics of *Origanum majorana* essential oil.

20 °C—density	0.894
15 °C—density	0.898
20 °C—refractive index	1.473
20 °C—optical rotation	+22.5 °C
80%—miscibility	1 mL of alcohol/ 1 volulme d’HE
SETAFLASH—flaspoint	55.9 °C

**Table 2 biomimetics-09-00715-t002:** Pesticides: dosage par GC MS detected XSD.

Pesticides: Dosage Par GC MS Detected XSD (Method Multiresilides #NF V03-110) List of Pesticides Researched (European Pharmacopea):	Results
Alachlor, Aldrine, BromophosEthyl, BromphosMethyl, Chlordane, Chlorfenvinphos, Chlorpyriphos, ChlorpyriphosMethyl, ChlorthalDimethyl, Cyfluthrine, Cyhalothrine lambda, Cypermethrine, Dichlofluanide, Dichlorvos, Dicofol (Ronnel), Fenchlorphos-oxon, Fenvalerate, Fluvalinate, Heptachlor, Heptachlorepoxixe, Hexachorobenzene, Hexachlorocyclohexane alpha, Hexachlorocyclohexane beta, Hexachlorocyclohexane delta, Hexachlorocyclohexane epsilon, Lindane, Methoxychlore, Mirex, Naled, o,p′-DDD, o,p′-DDE, o,p′-DDT, Oxychlordane, p,p′-DDD, p,p′-DDE, pp′-DDT, Pentachloroaniline, Pentachloroanisole, Permethrine, Phosalone, Procymidone, Profenophos, Prothiofos, Quintozene, S421, Tecnazene, Tetradifon, Vinclozoline.	<LMR * * Limit maximal residual

**Table 3 biomimetics-09-00715-t003:** Study groups (number of patients and salivary samples).

	*n* Patients	Salivary
Bruxismitle patients with *Origanum majorana* odor during 21 NeurOptimal-NO-sessions	12	120
Bruxismitle patients trained during 21 NeurOptimal sessions, without *Origanum majorana* essential oil	12	120
Bruxismitle patients (untrained in NO and without *Origanum majorana* exposure)	20	150
Controls (without bruxism and non-trained in NeurOptimal)	30	100
Controls expose to *Origanum majorana* odor without NO training	10	50
Controls trained in NeurOptimal	5	50
Total samples	89	590

**Table 4 biomimetics-09-00715-t004:** Times of Zen modes.

Session	Zen 1	Zen 2	Zen 3	Zen 4	Total Time
Demo	5	5		5	15
First	11	11		11.5	33.5
Second	8	12	5	8.5	33.5
Third	7	8	10	8.5	33.5
Regular	5	7	14	7.5	33.5
Extended	5	10	20	10	45.0

**Table 5 biomimetics-09-00715-t005:** Modes of NeurOptimal learning are indicated as follows.

Modes	Exercise	Difference Between Modes
Zen 1	Warm up	Both sides of the brain are trained separately over all frequencies, no comparison between sides
Zen 2	Weight lift	Both sides of the brain are trained together within each separate target range
Zen 3	Endurance	Both sides of the brctrain are trained together over all frequencies at the same time
Zen 4	Cool down	Integration of learning

**Table 6 biomimetics-09-00715-t006:** Effects of *Origanum majorana* essential oil on salivary cortisol levels.

Pre vs. Post	Pre vs. Pre-AE	Pre-AE vs. Post	Pre-AE vs. Post-AE	Post vs. Post-AE
*p* = 0.049 (S12)	*p* = 0.037 (S12)	*p* = 0.024 (S6)	*p* = 0.049 (S1) (S21) *p* = 0.14, n.s	*p* = 0.014 (S1) *p* = 0.1 (H2), n.s

## Data Availability

Original contributions are included in the article, further inquiries can be directed to the corresponding author (José Joaquín Merino).

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
