# Peer review of "Neurofeedback Technology Reduces Cortisol Levels in Bruxismitle Patients: Assessment of Cerebral Activity and Anxiolytic Effects of Origanum majorana Essential Oil"

_biomimetics, 2024, doi:10.3390/biomimetics9110715_

Round 1

Reviewer 1 Report

Comments and Suggestions for Authors

The study investigates the impact of neurofeedback technology, particularly the NeurOptimal (NO) system, on bruxistic patients' stress levels, focusing on the use of Origanum majorana essential oil.

Limited Sample Size:

Despite the large number of salivary samples, the number of participantsis relatively small for drawing generalized conclusions. The findings, while promising, might not be broadly applicable to all bruxistic patients.

Ambiguity in Data Interpretation:

The concept of "Divergence" as a measure of brain activity is somewhat ambiguous. While it is presented as an indicator of neural efficiency, the article does not clearly explain the biological mechanism underlying this concept. Additionally, the fluctuations in brain activity during sessions, especially around session 8, are not sufficiently clarified in the discussion.

Potential Confounding Variables:

The study excludes patients with several pre-existing conditions (e.g., diabetes, neurological disorders), but it does not account for factors like sleep patterns, diet, or other lifestyle variables that could impact cortisol levels or brain activity. This raises questions about the extent to which the reductions in cortisol can be solely attributed to the NO training and essential oil.

Overemphasis on Statistical Significance:

While the authors emphasize the significance of their statistical results, certain p-values hover close to the threshold of significance (p=0.05), particularly in relation to divergence values. This suggests that while there are trends, they may not be as robust as the paper suggests.

Lack of Long-term Follow-up:

The study concludes after 21 sessions of NO training, but it does not investigate whether the observed reductions in cortisol and brain activity are sustained over the long term. Given the chronic nature of stress and bruxism, this could limit the practical applicability of the findings.

Author Response

(x)

( )

Comments and Suggestions for Authors

The study investigates the impact of neurofeedback technology, particularly the NeurOptimal (NO) system, on bruxistic patients' stress levels, focusing on the use of Origanum majorana essential oil.

Limited Sample Size:

Despite the large number of salivary samples, the number of participants is relatively small for drawing generalized conclusions. The findings, while promising, might not be broadly applicable to all bruxistic patients.

Dear reviewer

Thanks for all your comments, which help us to improve the R1 version.

Cortisol levels were measured in 89 participants and 590 salivary saples) by ELISA in salivary samples. This mean 12 of these 89 participants are bruxistic patients. There was two groups of bruxistic participants who smell Origanum majorana essential oil during each NeurOptimal training (21 sessions), and other group of bruxistic participants (n=12) without smelling this odor during each NO session (21 sessions); the study group also enrolled controls without bruxism, and stressed participants without bruxism (n=12). Thus, we enrolled a total number of 89 participants.

With regard to cortisol levels, a recent metaanalysis by Lu et al confirmed that cortisol is a good predictor of bruxism in patients. A systematic review confirmed the good efficacy of cortisol for the identification of bruxism in subjects. In fact, the association between high salivary cortisol upon waking and bruxism has been confirmed in several studies (AlSahman L, et al. 2024. Is There a Relationship between Salivary Cortisol and Temporomandibular Disorder: A Systematic Review. Diagnostics (Basel). [2024], 5;14(13):1435). The possible relationship between bruxism and elevated salivary cortisol levels associated with the etiology of anxiety, fear, frustration, stress and also bruxism have been confirmed in several published studied (Fritzen VM, et al. 2022. Levels of salivary cortisol in adults and children with bruxism diagnosis. A systematic review and meta-analysis. J Evid Based Dent Pract (2022), 22(1):101634; Suprajith T, et al. 2022. Effect of Temporomandibular Disorders on Cortisol Concentration in the Body and Treatment with Occlusal Equilibrium. J Pharm Bioallied Sci. (2022), 14(Suppl 1):S483-S485).Thus, the possible contribution of emotional altered responses in bruxism should be not excluded. Including in our study, since several studies agree with cortisol salivary elevations as possible biomarkers of bruxism (AlSahman L, et al.2024); in addition, NeurOptimal training is a safe neurothecnology able to promote anxyolitic effects in patients by decreasing salivary cortisol levels in conjunction with Origanum majorana essential oil during repeated NO overtraining in bruxistic participants, which were not observed in participants without bruxism in our study.In these studies, the search strategy was developed using the following terms: "cortisol", "bruxism" and "sleep bruxism¨ indicated several database until May 2020 (Lu et al., 2020). These authors evaluate six primary studies, involving 854 participants, who met the criteria; the confirmed that cortisol is a good marker that identify possible difference in bruxism as compare to controls. In fact, despite differences in the methodologies adopted for the collection of data and analysis of salivary levels, this metanalysis found significant higher salivary cortisol in adult patients with bruxism (Lu et al., 2020. Thus, they confirmed the validation of salivary cortisol levels, which agree with our findings in bruxistic participants (Lu, et al. 2022. Salivary cortisol levels and temporomandibular disorders—A systematic review and meta-analysis of 13 case-control studies. Trop. J. Pharm. Res. 2022, 21, 1341–1349).

The survey identified a total of 105 studies, but 13 of these studies were evaluated. Of these, 7 studies were excluded because they did not present a control group. In adult participants, bruxistic patients  showed significant higher salivary cortisol levels as compare to controls. The conclussion of this study confirmed a positive association between bruxism and higher levels of salivary cortisol upon waking, it brings a possible relationship between altered emotional stress responses in bruxistic adults (Lu, L, et al. 2020. Salivary cortisol levels and temporomandibular disorders—A systematic review and meta-analysis of 13 case-control studies. Trop. J. Pharm. Res. 2022, 21, 1341–1349; Gameiro GH, et al. How may stressful experiences contribute to the development of temporomandibular disorders? Clin Oral Investig. (2006), 10(4):261-8; Fritzen VM, et al. LEVELS OF SALIVARY CORTISOL IN ADULTS AND CHILDREN WITH BRUXISM DIAGNOSIS: A SYSTEMATIC REVIEW AND META-ANALYSIS. J Evid Based Dent Pract. 2022 Mar;22(1):101634).Our findings agree with these findings since we measured salivary cortisol levels at similar hours (8 A.M to 10 A.M).There is a significant impact of psychosocial stress on the etiopathogenesis of TMD (Suprajith, T.; et al. Effect of Temporomandibular Disorders on Cortisol Concentration in the Body and Treatment with Occlusal Equilibrium. J. Pharm. Bioallied Sci. 2022, 14, S483–S485; D’Avilla, B.M.; et al. Comorbidity of TMD and malocclusion: Impacts on quality of life, masticatory capacity and emotional features. Braz. J. Oral Sci. 2019, 18, e191679).

In our study, the figure-1 indicates divergences (diferences of cerebral activities at prelanring sessions minus postraining values) in bruxistic patients with/without Origanum majorana stimulation during repeated NO overtraining (21 sessions). These brain activities were assessed in 12 bruxistic participants without Origanum majorana stimulation (252 measurements of cerebral activity) and also 12 bruxistic participants that inhalates this fragrance (252 too; total: 504).

Ambiguity in Data Interpretation:

¨The concept of "Divergence" as a measure of brain activity is somewhat ambiguous. While it is presented as an indicator of neural efficiency, the article does not clearly explain the biological mechanism underlying this concept. Additionally, the fluctuations in brain activity during sessions, especially around session 8, are not sufficiently clarified in the discussion¨.

We must take into account that fluctuations in cerebral activities are attributable to stress or normal changes on cerebral activity under not calm states in patients, which could occur during repeated NeurOptimal training sessions. However, these observed and signifficant changes on session-8 reached increased on divergence at session-8 could be justify by distraction and lack of concentration during NeurOptimal sessions or symply are part of the NO training, meaning that the calm state has not been reached at this session-8. In fact, the calm state was only observed after concludiing all 21 NeurOptimal training sessions.It is possible that a calm state need repeated overtraining and relaxation state, whihc are only reached by NO overtraining.

Pruessner JC et al reported that neither age, weight, nor smoking affect  the baseline or peak cortisol levels. Sleep duration, time of awakening and alcohol consumption also appeared to be unrelated to early morning free cortisol levels. However, patients taking alcohol were excluded in our study. These authors conclude that early morning cortisol levels can be a reliable biological marker for the individual's adrenocortical activity when measured repeatedly with strict reference to the time of awakening as we evaluated in our study. In fact, salivary cortisol levels were measured by ELISA from 8 A.M to 10 A.M. The observed increase after awakening is consistent, showed good intraindividual stability across days and weeks (Pruessner JC, Wolf OT, Hellhammer DH, Buske-Kirschbaum A, von Auer K, Jobst S, Kaspers F, Kirschbaum C. Free cortisol levels after awakening: a reliable biological marker for the assessment of adrenocortical activity. Life Sci. 1997;61(26):2539-49), This study confirmed the convenience of salivary cortisol levels at times of determination in our study.

¨Additionally, the fluctuations in brain activity during sessions, especially around session 8, are not sufficiently clarified in the discussion¨.

The fluctuations on cerebral activities (divergences) are really manifestation of temporal unpleasant effects during certain training sessions, in this case at session-8. In fact, brain activities (DIV) increased at session-8, reflecting some specific unpleasant effects without toxicity of Origanum majorana essential oil in patients who smell this fragrance during NO overtraining (see session 8: Figure 3 a,b); this evidence agrees with some reported unpleasant effects of some essential oils as isovarelic acid from Rosmarinus offiinales essential oil in  healthy participants expose to this odor in the environment but without using nasal filters (49,50,9). These fluctuations on cerebral activities suggest the calm state is not reached yet. Collectively, these peaks of cerebral activities are necessary to discard unpleasant odors as occurs at NO session-8 in bruxistic participants. When divergences (DIV) at post-training are negative or close to pre-learning values, this mean bruxistic patients are close to a calm or relaxation state as consequence of anxiolytic effects of NO overtraining.Based on the time of occurrence (circadian manifestations), two types of bruxism have been described thus far: i) Sleep bruxism, which is characterized by rhythmic masticatory muscle activity and occasional grinding; and ii) awake bruxism, characterized only by a clenching or bracing-type activity (Manfredini D, et al. Epidemiology of bruxism in adults: A systematic review of the literature. J Orofac Pain. 2013;27:99–110).

Bruxism is defined as a repetitive jaw muscle activity characterized by clenching or grinding of the teeth and/or bracing or thrusting of the mandible; however, it is not regarded as a movement disorder or a sleep disorder in otherwise healthy individuals- Sleep bruxism, which is characterized by rhythmic masticatory muscle activity and occasional grinding; and ii) awake bruxism, characterized only by a clenching or bracing-type activity. Several studies support a different etiology among the two types of bruxism. However, Manfredini et al, reported the prevalence of awake bruxism ranged from 22 to 31%, whereas the prevalence of sleep bruxism was 12,8%. In particular, when the patient have increased levels of emotional stress, this could lead to an increase in head and neck muscle tonicity, although it could also lead to an increase in the level of non-functional muscle activity, such as bruxism or tooth clenching (16). Furthermore, the sympathetic activity or tone may also be influenced by emotional stress  as etiological factor that can influence temporomandibular disorder (TMD) symptoms. In addition, the involvement of stress in orofacial musculature modulation has been also suggested in several studies (Fillingim RB, et al. Psychological factors associated with development of TMD: The OPPERA prospective cohort study. J Pain. 2013;14 (12 Suppl):T75–T90).

Potential Confounding Variables:

The study excludes patients with several pre-existing conditions (e.g., diabetes, neurological disorders), but it does not account for factors like sleep patterns, diet, or other lifestyle variables that could impact cortisol levels or brain activity. This raises questions about the extent to which the reductions in cortisol can be solely attributed to the NO training and essential oil.

It is true that our selected patients have sleep problems although we don`t have good tools for its identification. Even so, we know that slepp problems are also consequence of bruxism and cortisol levels could also fluctuate at different NeurOptimal sessions. However, our findings clearly showed anxiolytic effects by decreasing salivary cortisol levels by NeurOptimal overtraining after 21 sessions in bruxistic participants while Origanum majorana odor expose during NO could partially contribute to these anxiolytic effects in our study.

The life style in participants is difficult to monitorize by behavioral task and the effects of diet and other confunder factors are not monitorized in this study. However, some confunders factors are excluded in the design of this study, thus suggesting specific effects on our evaluated variables (cortisol and divergences as index of cerebral activity).

We agree with you that sleep could affect cerebral activities in bruxistic participants. However, the analysis of cerebral activities fail to detect of significant change after 21 NO sessions on cerebral activities in bruxistic patients without/with Origanum majorana exposure during each NO session. Our results confirmed that NeurOptimal overtraining after 21 sessions really reduces cortisol levels in bruxistic participants, and sleep problems are present in these participants. Based on the available evidence, Porporatti AL et al in a metanalysis study confirmed that stress factors in questionnaire-based studies, and higher salivary cortisol levels are biomarkers of bruxism. With regard to style factors and its influence on bruxism, although odds ratios greater than unity indicate an association, this was not high for carbonated beverages and many other acidic foods or drinks. (Alam MK, et al. Salivary Biomarkers and Temporomandibular Disorders: A Systematic Review conducted according to PRISMA guidelines and the Cochrane Handbook for Systematic Reviews of Interventions. J Oral Rehabil. 2024 Feb;51(2):416-426). Thus, bruxism and cerebral activities seems not be affected by diet factors.

Examining at fourteen joung participants years may not be ideal, as the determinants of erosion/tooth wear have not acted for long, the indices do not discriminate sufficiently and proportionately few subjects have dentine exposed on smooth surfaces. Although this study was performed in adolescent and our study evaluated adults, the contribution of diet factors is not strong in the manifestation of bruxism (Milosevic A, et al. Epidemiological studies of tooth wear and dental erosion in 14-year old children in North West England. Part 2: The association of diet and habits. Br Dent J. 2004 Oct 23;197(8):479-83; discussion 473; qui 505), which is really associated withs stress and psychological factors (Kuhn M, Türp JC. Risk factors for bruxism. Swiss Dent J. 2018 Feb 12;128(2):118-124). In fact, mental stress stimulate the hypothalamic–pituitary–adrenal (HPA) axis, which triggers cortisol release from the adrenal cortex and is considered a marker of stress and anxiety (AlSahman L, et al. Is There a Relationship between Salivary Cortisol and Temporomandibular Disorder: A Systematic Review. Diagnostics (Basel). 2024 Jul 5;14(13):1435). These psychological factors, especially stress, are considered one of the main areas of temporal mandibule Temporomandibular disorders (TMDs)  aetiology, but their role in the occurrence of TMD is inconclusive (Atsu, S.S.; et al. Oral parafunctions, personality traits, anxiety and their association with signs and symptoms of temporomandibular disorders in the adolescents. Afr. Health Sci. 2019, 19, 1801–1810). Although a good diet could induce antiinflammatory effects, the inclussion of appropiate controls for cortisol comparisons in our study, exclude antiinflammatory cortisol dependent effects in our study. Given their moderate-high sociocultural state, diet and nutritional status seems to be within normal standars; further studies will evaluate the influence of diet factors in bruxism. Additionally, Origanum majorana essential fragance exposure during each 21 NeurOptimal training sessions, contains compounts with antioxidant, antiinflammatory or anxiolytic effects, which did not signifficantly affect brain activities after concluding all 21 repeated NO overtraining in patients. In additon, tehir cerebral activities could not be affected by their sociocultural status in our study.

However, cortisol levels is a good marker for assessment of HPA axis activity (Cui, Q et al. Morning Serum Cortisol as a Predictor for the HPA Axis Recovery in Cushing’s Disease. Int. J. Endocrinol. 2021, 2021, 4586229]. In fact, pain and stress lead to parafunctional habits [Ohrbach, R.; Michelotti, A. The Role of Stress in the Etiology of Oral Parafunction and Myofascial Pain. Oral Maxillofac. Surg. Clin. 2018, 30, 369–379], like bruxism are linked with signs and symptoms of Temporal Mandibullar Disease [Atsu, S.S. et al. Oral parafunctions, personality traits, anxiety and their association with signs and symptoms of temporomandibular disorders in the adolescents. Afr. Health Sci. 2019, 19, 1801–1810]. Several studies have shown a correlation between HPA axis dysregulation and TMD (Gameiro, G.H.; et al. How may stressful experiences contribute to the development of temporomandibular disorders? Clin. Oral Investig. 2006, 10, 261–268; Cui, Q.; et al.. Morning Serum Cortisol as a Predictor for the HPA Axis Recovery in Cushing’s Disease. Int. J. Endocrinol. 2021, 2021, 4586229).

Additionally, monitoring cortisol levels over time can provide valuable information about the effectiveness of stress management techniques in reducing TMD symptoms AlSahman L, et al. Is There a Relationship between Salivary Cortisol and Temporomandibular Disorder: A Systematic Review. Diagnostics (Basel). 2024 Jul 5;14(13):1435), as we have observed with our study with NeurOptimal technology in bruxistic participants. Two randomized control trials showed reductions in salivary cortisol levels after treatment in patients with TMD compared to those treated with a placebo (Goyal et al Salivary Cortisol Could Be a Promising Tool in the Diagnosis of Temporomandibular Disorders Associated with Psychological Factors. J. Indian Acad. Oral Med. Radiol. 2021, 32, 354–359;  Magri, L.V.; et al. Non-specific effects and clusters of women with painful TMD responders and non-responders to LLLT: Double-blind randomized clinical trial. Lasers Med. Sci. 2018, 33, 385–392). In addition, several randomized control trials confirmed a positive association between higher cortisol levels and TMD (Rosar, et al. Effect of interocclusal appliance on bite force, sleep quality, salivary cortisol levels and signs and symptoms of temporomandibular dysfunction in adults with sleep bruxism. Arch. Oral Biol. 2017, 82, 62–70; Magri, L.V.; et al., 2021). In these studies, their salivary cortisol levels were quantified at morning as we did in our study: Several etiological factors—mainly stress, anxiety, and depression- can explain why cortisol levels are high in bruxism. Recent studies have shown a positive association between stress, anxiety, and depression and the occurrence of TMD (Anna, S et al. The influence of emotional state on the masticatory muscles function in the group of young healthy adults. Biomed. Res. Int. 20152015, 174013). These evidences agree with the anxiolytic effects of NeurOptimal training in our study.

Overemphasis on Statistical Significance:

While the authors emphasize the significance of their statistical results, certain p-values hover close to the threshold of significance (p=0.05), particularly in relation to divergence values. This suggests that while there are trends, they may not be as robust as the paper suggests.

The repeated ANOVA does not show a significant effect for divergences (cerebral activities) after concluding 21 NeurOptimal training sessions ([F 487 (1,20)=1,27; p=0,2; n.s, Etha square=0,14, podwer: 0,85) by GreenhouseGeisser analysis (Figure 1a, Figure 3b).  This mean cerebral activities are not affected by NeurOptimal overtraining (21 sessions). Although cerebral activities are not regulated by NO overtraining, their cortisol levels were significantly reduced after 21 NO training sessions and Origanum majorana partially contribute to this anxiolytic effects. As a whole, our finfings suggest that Origanum majorana essential oil did not affect divergences (Divergence) after 21 NO training sessions but also could enhance anxiolytic effects of NeurOptimal overtraining in bruxistic participants.

In addition, please notice the study group also included controls and bruxistic participants without Origanum majorana essential oil exposure and without NeurOptimal training, which means the total number of patients is 89 with 590 salivary samples. We must also consider that NeurOptimal training is completed within 3 months after its initial training session.

Lack of Long-term Follow-up:

The study concludes after 21 sessions of NO training, but it does not investigate whether the observed reductions in cortisol and brain activity are sustained over the long term. Given the chronic nature of stress and bruxism, this could limit the practical applicability of the findings.

Yes, we indirectly investigated the possibel short/long-term reduction on salivary cortisol levles and/or brain activities (divergences) by sustained overtraining in NeurOptimal: In fact, NO overtraining was completed after 21 sessons and this procedure takes 90 days (3 months) in our study. In fact, two NeurOptimal training sessions takes by week in each patient, which means we need 11 weekd for concluding all NO training session, which equivales to 90 days (3 months). Thus, there is an indirect investigation are sustained over the long term in cortisol and brain activity in our study.

Cortisol levels were measured in 89 participants (590 salivary samples) by ELISA. The repeated ANOVA signifficantly decreased cortisol levels when we evaluated at sessions 1,6,12,18, 21 (S-1, 529 S6, S12, S18, S21) as a consequence of NO overtraining [F (1,4)=3,6; p<0.05). However, the Origanum majorana essential oil almost leads to reduce cortisol levels, reaching a trend.

On the other hand, long term effect of treatments in bruxisms are studied the literature since urgent solutions against bruxism are needed. Thus, there is not available results for comparison with our findings.

Thanks for your important comments again, which help us to improve our R1 manuscript version.

Reviewer 2 Report

Comments and Suggestions for Authors

The paper investigates the effects of NeurOptimal training and Origanum majorana essential oil inhalation on cortisol levels and brain activity in patients with bruxism. Below are a few comments:

1. Line number 119, 130 - if you are using 24 hr clock AM is not required.

2. The quality of the pictures needs improvement mainly the figures in page 15 and 16.

3. From the methodology the EEG electrode locations are C3 and C4, a 10/20 figure highlighting the location would be useful in section 2.5.2.2.

4. Section 2.5.2.4, needs further clarity on how the hemisphere separation was performed and the frequencies.

Author Response

Comments and Suggestions for Authors

The paper investigates the effects of NeurOptimal training and Origanum majorana essential oil inhalation on cortisol levels and brain activity in patients with bruxism. Below are a few comments:

1.Line number 119, 130 - if you are using 24 hr clock AM is not required.

Dear reviewer

Thanks for all your comments, which help us to improve this manuscript version.

You are rigth but it is important to precise the exactly moment for salivary cortisol isolation because cortisol at evening are lower as compare to morning values, which are the selected times for cortisol evaluation by ELISA. This is the reason by which we included 8-10 A,M becuase cortial at evening are not representative of NeurOptimal performmance evaluation.

  1. The quality of the pictures needs improvement mainly the figures in page 15 and 16.

All pictures have been improved in this R2 version

  1. From the methodology the EEG electrode locations are C3 and C4, a 10/20 figure highlighting the location would be useful in section 2.5.2.2.

The participant hookup consists of silver electrodes, applied to the patient’s ears & scalp, centered between the ear and the crown of the head on the bony ridge (Central points C3 & C4). The electrodes are applied with EEG paste. It is water soluble electrical conductance material composed primarily of salts that enhances the monitoring of the minute electrical pulses.

Firstable, separate left sensores from the right. Squeeze paste onto sensors without touching the paste with your fingers.Apply paste in the same way to the bacl sensors, then gently sequeeze them in the same way, to the bottom of the rigth ear (where the lobe joints the ear). Place the sensor on the cleared area by paste side down.

  1. Section 2.5.2.4, needs further clarity on how the hemisphere separation was performed and the frequencies.

Self-Optimization principle  of NeurOptimal is bases on the brain inherently runs a self-optimizing program that is constantly pruning neuronal connections that are not in use and adding new neuronal connections and hubs where required.

Neurofeedback plugs into this self-optimization program, giving the brain the ability to learn quickly and effortlessly.  The NeurOptimal ® Neurofeedback system recognizes the unconscious mind learns at a much faster rate than the conscious mind can work. This is a very different than other neurofeedback or biofeedback systems, which require participants to consciously learn to control their brainwave activity in order to benefit from the feedback.

The brain wave balance of Neurofeedback provides the brain with a moment-by-moment picture of how it is operating, mirroring it back to the brain. The feedback, which is a small break in the music, allows the brain to identify areas of wave frequency instability or “stress”, thus using this information to begin the self-optimization process.  Over time, the frequencies remain regulated and the brain and nervous system are measurably different than before the training.

An EEG (electroencephalogram) is a recording of the electrical activity of the brain. Brain wave frequencies are measured in Hertz, which is defined as wave cycles per second. Brain frequencies range from the lowest and slowest Delta waves, to Theta waves, to Alpha waves, to Beta waves, to the highest and fastest Gamma waves above 38 Hertz. Bandwidths of brain frequencies are associated with different states of consciousness.

The participant hookup consists of silver electrodes, applied to the patient’s ears & scalp, centered between the ear and the crown of the head on the bony ridge (Central points C3 & C4). The electrodes are applied with EEG paste. It is water soluble electrical conductance material composed primarily of salts that enhances the monitoring of the minute electrical pulses of the brain (www.zengarinstitute.com). (

Z-amp™ Amplifies Signals The electrode sensors pick-up the brain’s electrical signal and send that signal down a conductance wire to the Zengar Z-amp™. The Z-amp™ filters line noise and amplifies the brain wave signal. Other neurofeedback data looks “smeared” in comparison NeurOptimal’s ®  data, due to a sampling rate of 256 samples per second, coupled with incredible precision of filtering, targeting, and triggering of feedback.  At no time are electrical signals fed back to the brain.

Signal Separation into Frequencies and intensities left and right brain wave signals are separated by the computer software into their component frequencies and intensities. Intensity, for these purposes, is defined as a measure of the amount of electrical signal generated. This continuous data set is analyzed over time using non-linear statistics in order to determine when the brain and nervous system enter into a period of “unstable” operation.  Unstable periods are identified within milliseconds and feedback is given in the form of a pause in the music.Feedback is determined by continuously tracking the 3 variables of time, frequency, and intensity. This is called Joint time frequency analysis (JTFA).

Zen Modes The NeurOptimal® software automatically moves through four different modes called Zen 1,2, 3, & 4. These modes are similar to a warm up stretch, a weight training challenge, a cardiovascular endurance, and a cool down period to integrate the learning.

The brain wave has been broken down into its component frequencies from 1 Hz at the botton in NeurOptimal to  42 Hz at the top. In fact, NeurOptimal training must reach to lower frequencies as small as posible. The applied program was zengar 3 that is specific for both sides of the brain are trained together over all frequencies at the same time with endurance.

Thanks again for your valuable comments

.

Reviewer 3 Report

Comments and Suggestions for Authors

The author had used majorana (aromatherapy) essential oil to reduce stress and alertness states of the patients which was moniotored by Neurofeedback technology, however the authors need to quantify the stress level % in terms of salivary cortisol levels as with the participated patients. Methodology selected is well commented with the data presentations, where the qunatification is more important when coming to stress reduction in compare with other means. 

Author Response

The author had used majorana (aromatherapy) essential oil to reduce stress and alertness states of the patients which was moniotored by Neurofeedback technology, however the authors need to quantify the stress level % in terms of salivary cortisol levels as with the participated patients. Methodology selected is well commented with the data presentations, where the qunatification is more important when coming to stress reduction in compare with other means.

Firstable, we thanks all your valuable comments, which help us to improve this R1 version.

We have added a figure with the percentage of cortisol level as compare to controls. Please, notice salivary cortisol levels have been quantified all the time by ng/ml in salivary samples by interpolation within the standard curve with a commertial ELISA kit. The procedure of cortisol quantification is the same that we followed in a study with curcumin and cortisol in patients (Merino JJ, Parmigiani-Cabaña JM, Parmigiani-Izquierdo JM, Fernández-García R, Cabaña-Muñoz ME. Decreased Systemic Monocyte Colony Protein-1 (MCP-1) Levels and Reduced sCD14 Levels in Curcumin-Treated Patients with Moderate Anxiety: A Pilot Study. Antioxidants (Basel). 2024 Aug 29;13(9):1052. doi: 10.3390/antiox13091052. PMID: 39334711; PMCID: PMC11429384)
